# Ultrasound Examination for Cement Extrusion After Uni-Compartmental Knee Replacement

**DOI:** 10.3390/diagnostics15010112

**Published:** 2025-01-05

**Authors:** Peter Kam-To Siu, Wei-Ting Wu, Levent Özçakar, Ke-Vin Chang

**Affiliations:** 1Department of Orthopaedics and Traumatology, Queen Mary Hospital, Hong Kong; petersiuhere@gmail.com; 2Department of Orthopaedics and Traumatology, The University of Hong Kong, Hong Kong; 3Department of Physical Medicine and Rehabilitation, National Taiwan University Hospital, Bei-Hu Branch, Taipei 10845, Taiwan; wwtaustin@yahoo.com.tw; 4Department of Physical Medicine and Rehabilitation, College of Medicine, National Taiwan University, Taipei 100006, Taiwan; 5Department of Physical and Rehabilitation Medicine, Hacettepe University Medical School, Ankara 06100, Turkey; lozcakar@yahoo.com; 6Center for Regional Anesthesia and Pain Medicine, Wang-Fang Hospital, Taipei Medical University, Taipei 110301, Taiwan

**Keywords:** gonarthrosis, pain, arthroplasty, material, sonography

## Abstract

A 66-year-old woman presented with persistent knee effusion three months after undergoing a cemented medial uni-compartmental knee replacement. She was afebrile and able to walk with a stick. Physical examination revealed moderate effusion. Radiographs showed posteriorly extruded cement, while computed tomography confirmed the absence of implant loosening but was unable to adequately visualize the adjacent soft tissues due to metallic artifacts. Ultrasound identified posterior cement extrusion beyond the femoral component, causing a delamination tear of the posterior capsule and indentation on the medial gastrocnemius. Knee arthrocentesis yielded 60 mL of blood-stained fluid with unremarkable analysis, and the patient reported improvement following the procedure. To our knowledge, this is the first report to highlight the unique role of ultrasound in detailing the anatomy of extruded cement and its impact on adjacent soft tissues following knee replacement. We demonstrate the critical structures that should be evaluated and how ultrasound aids in managing this postoperative complication.

**Figure 1 diagnostics-15-00112-f001:**
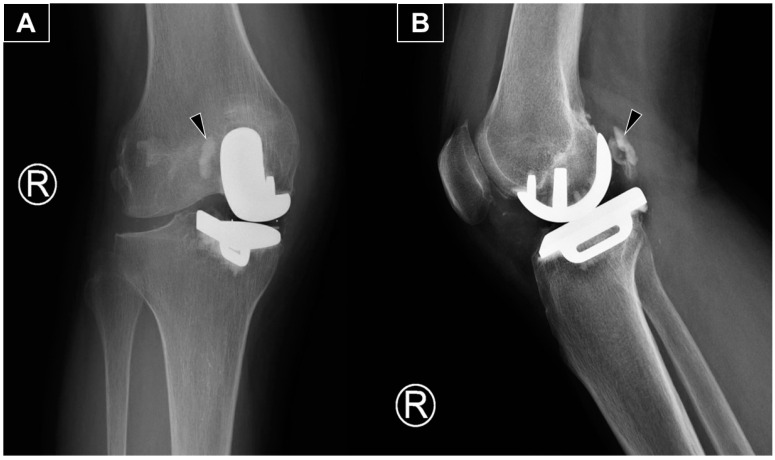
The anteroposterior (**A**) and lateral (**B**) radiographs reveal the main portion of the extruded cement (arrowhead) positioned posterolateral to the medial femoral metallic component. The radiograph is from a 66-year-old woman with isolated medial compartment osteoarthritis of the right knee who experienced mechanical knee pain, with a visual analog scale (VAS) score of 7/10. Her symptoms had limited her walking to 30 min with the assistance of a walking stick. She could ascend only one flight of stairs (using a stick and rail support) and was unable to squat due to knee pain and stiffness. After exhausting all non-operative treatments without significant improvement, she underwent a medial uni-compartmental knee replacement. She first presented to our clinic three months postoperatively with persistent knee swelling and posterior knee pain during deep flexion. She was afebrile and ambulated with a walking stick, though with some difficulty. Examination revealed moderate knee swelling with mild warmth and tenderness but no erythema. The surgical wound was well-healed. Active knee extension was complete, and flexion reached 100 degrees actively and 120 degrees passively, with posterior knee tenderness at the end range of flexion. Ligamentous stability was intact, and no posterior swelling or skin changes were observed. There was no calf swelling or other leg symptoms, and the neurovascular examination was normal. Radiographs showed an opacity posterior to the metallic femoral component, consistent with extruded cement from the fixation.

**Figure 2 diagnostics-15-00112-f002:**
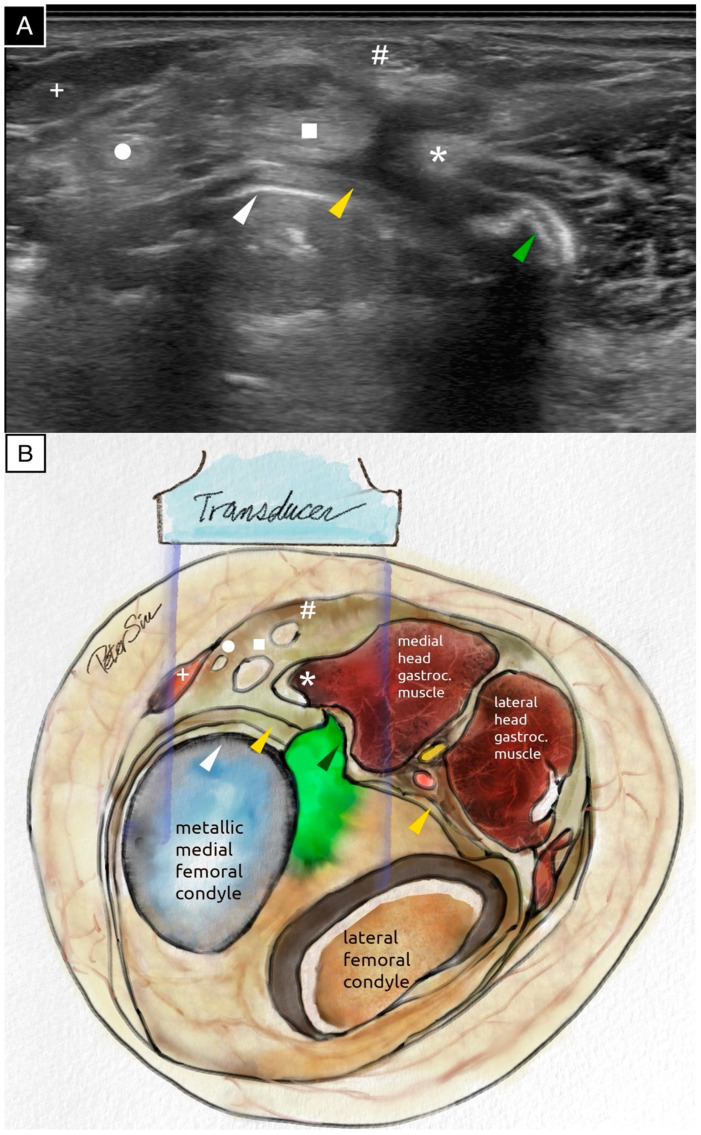
The transverse ultrasound scan (**A**) and the schematic illustration (**B**) of the posteromedial knee demonstrate extruded cement positioned posterior to the metallic femoral component. The extruded cement impinges on the posterior capsule and the medial head of the gastrocnemius muscle. The surface of the extruded cement reflects ultrasound waves, producing a surface echogenic line and casting a strong posterior acoustic shadow. The following anatomical structures are labeled: the extruded cement (green arrowhead), the posterior capsule (yellow arrowhead), the posterior border of the metallic femoral component (white arrowhead), the medial head of the gastrocnemius muscle (asterisk), the semimembranosus tendon (square), the gracilis tendon (circle), the semitendinosus tendon (hash sign), and the sartorius tendon (plus sign). The ultrasound image, acquired using the Fujifilm Sonosite PX with an L15-4 linear probe (operating at the general frequency setting, with the dynamic range set to 0 on a scale from −3 to +3, and the time gain compensation kept at the factory setting with no adjustments), provided a detailed assessment of the surrounding soft tissues and highlighted the extensive application of ultrasound technology in the medical field [1]. An ultrasound evaluation is provided in Appendix A.

**Figure 3 diagnostics-15-00112-f003:**
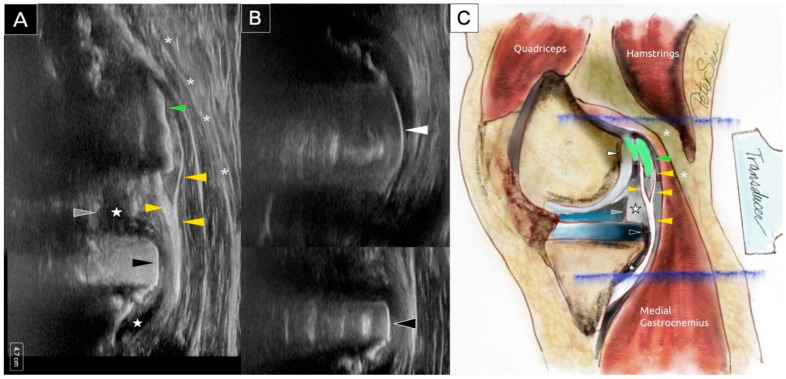
Longitudinal sonographic imaging of the posteromedial knee reveals extruded cement located posterior to the metallic femoral component, causing laceration of the posterior capsule and impingement on the medial head of the gastrocnemius muscle (**A**). In a more medial longitudinal scan, no extruded cement is visible, exposing the underlying metallic femoral component (**B**). Both the femoral and tibial metallic components produce prominent reverberation artifacts. The schematic illustration (**C**) corresponds to the findings in (**A**). The following anatomical structures are labeled: extruded cement (green arrowhead), posterior capsule (yellow arrowhead), polyethylene insert (gray arrowhead), metallic tibial tray (black arrowhead), posterior knee joint effusion (star), and medial head of the gastrocnemius muscle (asterisk). An ultrasound evaluation is provided in Appendix A.

**Figure 4 diagnostics-15-00112-f004:**
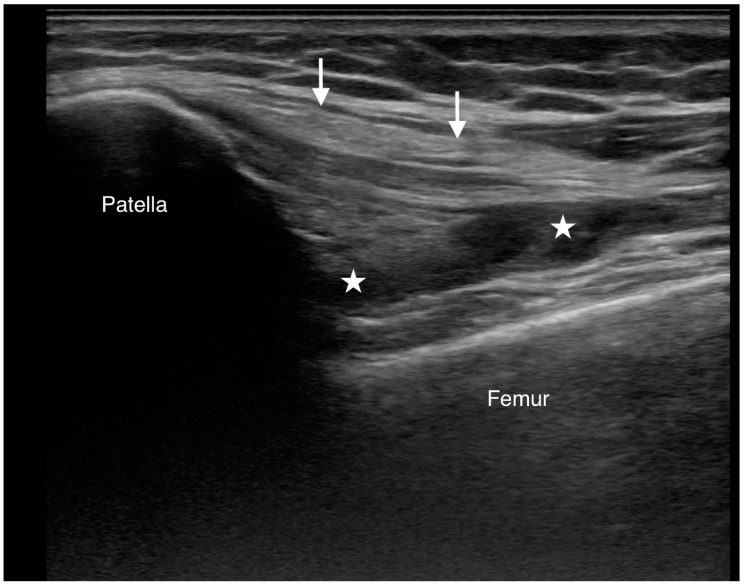
The extruded cement and capsular laceration appeared to contribute to significant knee effusion, as observed on ultrasound. A longitudinal ultrasound scan of the anterior knee demonstrates effusion in the suprapatellar pouch (star). The quadriceps tendon is indicated by a white arrow.

**Figure 5 diagnostics-15-00112-f005:**
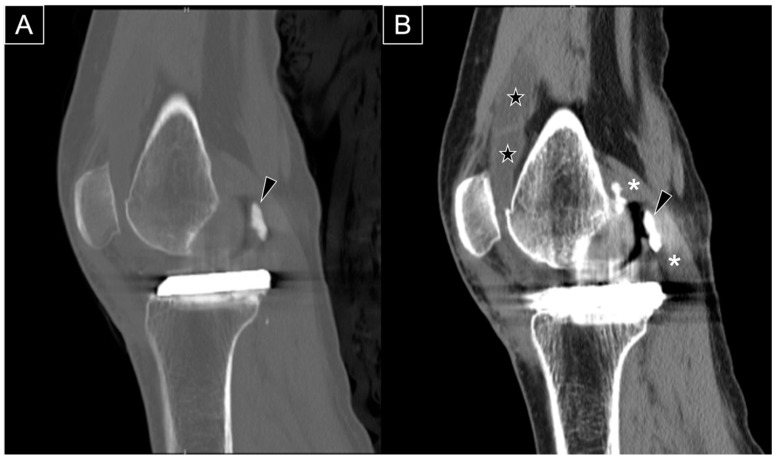
Sagittal computed tomography (CT) scans in the bone window (**A**) and soft tissue window (**B**) demonstrate extruded cement (black arrowhead) positioned posterior to the metallic femoral component. The CT scans confirm that both the femoral and tibial metallic components are securely fixed to the bone. The extruded cement impinges on the medial head of the gastrocnemius muscle (asterisk), and significant knee effusion is evident (star).

**Figure 6 diagnostics-15-00112-f006:**
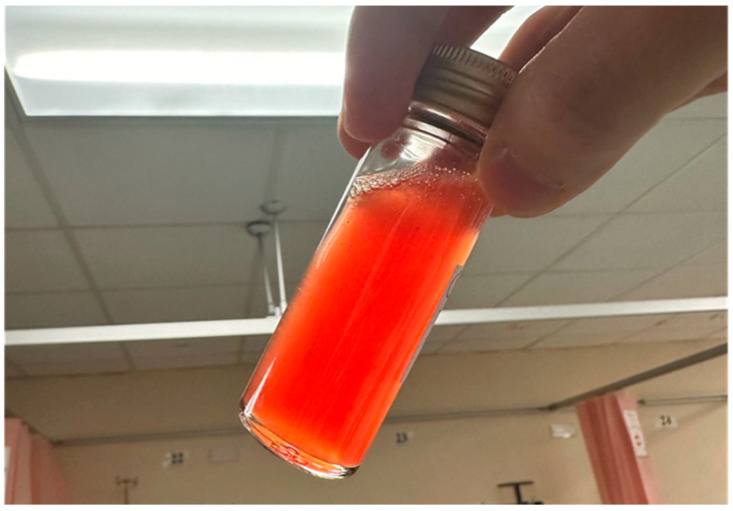
The patient subsequently underwent knee arthrocentesis, which yielded 60 mL of light blood-stained fluid, without any pus or fat globules. Joint fluid analysis revealed a total cell count of 2250, while cultures, crystal analysis, and cytology were all negative. Blood tests showed normal white blood cell count, C-reactive protein, and bone profile, including calcium, phosphate, and alkaline phosphatase levels. Following the procedure, the patient reported reduced knee pain and swelling, enabling her to walk independently. At three months post-arthrocentesis, there has been no re-accumulation of knee effusion, and her knee pain has significantly improved, allowing her to walk unaided.

**Figure 7 diagnostics-15-00112-f007:**
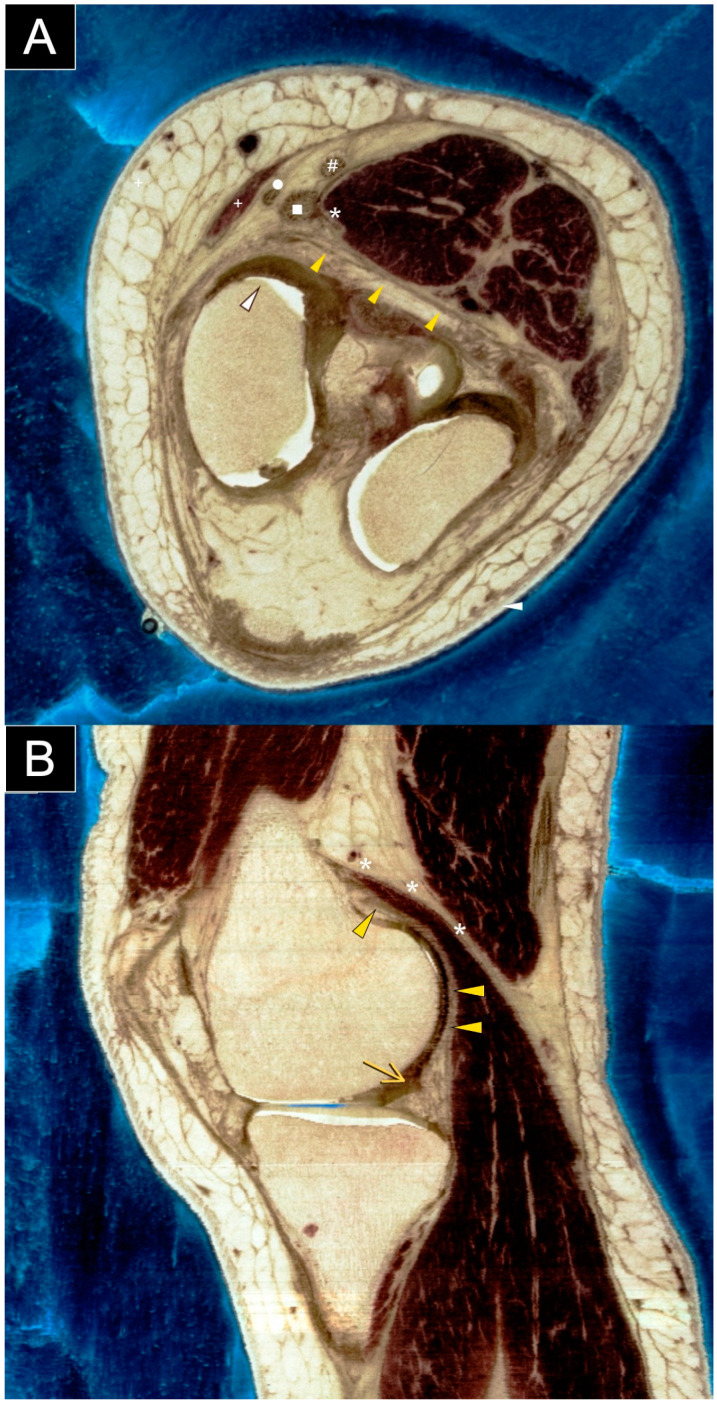
Axial (**A**) and sagittal (**B**) sections of a cadaveric knee specimen illustrate the structures located at or adjacent to the posteromedial knee capsule. Understanding the normal anatomy of the posteromedial knee is essential for evaluating and managing cases involving suspected cement extrusion following knee replacement surgery. The labeled structures include the posterior capsule (yellow arrowhead), posterior femoral condyle (white arrowhead), posterior horn of the medial meniscus (yellow arrow), medial head of the gastrocnemius (asterisk), semimembranosus tendon (square), gracilis tendon (circle), semitendinosus tendon (hash sign), and sartorius tendon (plus sign). Images are courtesy of the Visible Human Project^®^ of the National Library of Medicine. This article highlights cement extrusion following knee replacement, a phenomenon reported in the literature. A PubMed search using the keywords ‘knee replacement’, ‘cement’, and ’extrusion’ identified several reports. Otani et al. [2] described a 57-year-old patient with rheumatoid arthritis who developed posterolateral impingement after total knee arthroplasty. Radiographs revealed bone cement extruding posterolaterally to the tibial tray. Arthrotomy via a posterolateral approach showed that impingement was caused not only by cement extrusion against the fibular head but also by proximal tibiofibular joint instability. Palco et al. [3] and Kim et al. [4] reported cases of bone cement fragments causing mechanical symptoms after medial uni-compartmental knee replacement. Radiographs were obtained, and arthroscopic removal of the loose cement bodies resulted in positive outcomes. To our knowledge, this is the first report to highlight the unique role of ultrasound in detailing the anatomy of extruded cement and adjacent soft tissues after knee replacement surgery. Ultrasound is particularly valuable for assessing soft tissues around metallic implants, offering both anatomical and functional insights. This modality allows clinicians to evaluate and monitor soft tissue structures adjacent to metallic implants, guiding treatment strategies and potentially improving patient outcomes. However, while our article highlighted the usefulness and portability of ultrasound imaging in evaluating post-arthroplasty knee complications, its application is limited to a single case. Therefore, we have included a SWOT analysis (Appendix A) to support the broader application of ultrasound in the general population.

## Data Availability

All the ultrasound images, videos, and illustration drawings are produced and owned by the first author of this article P.K.-T.S. They are available upon reasonable request to him.

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
