# Peer review of "Ultrasound Examination for Cement Extrusion After Uni-Compartmental Knee Replacement"

_diagnostics, 2025, doi:10.3390/diagnostics15010112_

Round 1
Reviewer 1 Report
Comments and Suggestions for Authors
Please provide the scan parameters and if the work is repeatable.
Also mention if there could be any application of your method using MRgFUS or USgFUS by referring to the paper titled as 'Principles of Focused Ultrasound. Minim. Invasive Ther. Allied Technol. 2018, 27, 41–50'
Also please explain the merit of your study over one subject only? If this could be applied to larger populations. Provide SWOT analysis of your study
Author Response
Reviewer 1
Comment:
Please provide the scan parameters and if the work is repeatable.
Response:
The scan parameters were set using a Fujifilm Sonosite PX with an L15-4 linear probe, operating at the general frequency setting. The dynamic range was set to 0 (on a scale from -3 to +3), and the time gain compensation remained at the factory setting with no adjustments. These specifications are commonly available on basic ultrasound machines, ensuring the procedure's repeatability across different equipment.
Comment:
Also mention if there could be any application of your method using MRgFUS or USgFUS by referring to the paper titled as 'Principles of Focused Ultrasound. Minim. Invasive Ther. Allied Technol. 2018, 27, 41–50'
Response: We believe our article is relevant to the one suggested by the reviewer. The suggested article has been cited in the sentence: “The ultrasound image, acquired using the Fujifilm Sonosite PX with an L15-4 linear probe (operating at the general frequency setting. The dynamic range was set to 0 (on a scale from -3 to +3), and the time gain compensation remained at the factory setting with no adjustments.), provided a detailed assessment of the surrounding soft tissues and highlighted the extensive application of ultrasound technology in the medical field.”
Comment
Also please explain the merit of your study over one subject only? If this could be applied to larger populations. Provide SWOT analysis of your study
Response: We appreciate the reviewer’s suggestion. Cement extrusion is an uncommon occurrence in patients undergoing knee arthroplasty. Therefore, this exceptional case highlights the unique diagnostic benefit of ultrasound, a portable imaging tool, in screening for post-arthroplasty complications when patients present with painful symptoms. The SWOT analysis will be provided in the supplemental material as:
“Strengths
This case study introduces ultrasound as a novel tool for evaluating posterior cement extrusion and its effects on soft tissues after knee replacement. It highlights ultrasound's unique ability to provide detailed, non-invasive, radiation-free assessment of soft tissue injuries next to orthopedic implants (metal and cement in this case), which cannot be well delineated by other imaging modalities like computed tomography or magnetic resonance imaging. The study also serves as an educational resource, showing which structures to evaluate in postoperative care.
Weaknesses
The study is based on a single case, limiting its generalizability to broader patient populations. It lacks long-term outcome data and does not compare ultrasound to other imaging modalities, such as magnetic resonance imaging. However, it is well known that despite recent advances, it is inherent weakness of magnetic resonance imaging in delineating soft tissues immediately next to metallic implants, ie. the posterior knee capsule in this case.
Opportunities
The study supports the adoption of ultrasound in postoperative evaluations, particularly when traditional imaging is hindered by artifacts due to surgical implants (metals, cement etc.). It also opens the door for further research to validate ultrasound’s role in managing similar complications and for the development of specialized imaging protocols in orthopedics.
Threats
Ultrasound’s reliance on operator skill can lead to variability in results. Additionally, resistance from clinicians accustomed to traditional imaging methods and resource limitations in access to advanced ultrasound technology may limit ultrasound’s widespread use.
Reviewer 2 Report
Comments and Suggestions for Authors
This is a very educative and well-written case report. The images and supplemental videos are very illustrative. The anatomical details are excellent. I thank the authors for their work. I only recommend the authors to clarify the patient outcome. Previous cases in the literature reported that the extruded cement was removed arthroscopically. Therefore, a question arises for the recurrence of the symptoms for this patient. Is arthrocentesis sufficient for treatment?
Author Response
Reviewer 2
This is a very educative and well-written case report. The images and supplemental videos are very illustrative. The anatomical details are excellent. I thank the authors for their work. I only recommend the authors to clarify the patient outcome. Previous cases in the literature reported that the extruded cement was removed arthroscopically. Therefore, a question arises for the recurrence of the symptoms for this patient. Is arthrocentesis sufficient for treatment?
Response: We appreciate the reviewer’s kind suggestion. The patient was last seen recently, at three months post-arthrocentesis. There has been no re-accumulation of knee effusion, and her knee pain has significantly improved, allowing her to walk unaided. This information has been incorporated into the revised manuscript. Additionally, cement extrusion is heterogeneous and spans a spectrum of severity. When extruded cement fragments become loose bodies, they can cause mechanical symptoms that require removal through arthroscopic or open procedures, as reported by Palco et al. and Kim et al. Conversely, in less severe cases, cement extrusion may not require surgical removal, as noted by Elmadag et al. Our patient has shown significant improvement with non-operative treatment, making surgical removal unnecessary.
Round 2
Reviewer 1 Report
Comments and Suggestions for Authors
it is acceptable